# Functional Drug Screening of Small Molecule Inhibitors of Epigenetic Modifiers in Refractory AML Patients

**DOI:** 10.3390/cancers14174094

**Published:** 2022-08-24

**Authors:** Jessica L. Dennison, Hassan Al-Ali, Claude-Henry Volmar, Shaun Brothers, Justin Watts, Claes Wahlestedt, Ines Lohse

**Affiliations:** 1Center for Therapeutic Innovation, Miller School of Medicine, University of Miami, Miami, FL 33136, USA; 2Department of Psychiatry and Behavioral Sciences, Miller School of Medicine, University of Miami, Miami, FL 33136, USA; 3Miami Project to Cure Paralysis, Department of Neurological Surgery, Miller School of Medicine, University of Miami, Miami, FL 33136, USA; 4Peggy and Harold Katz Family Drug Discovery Center, Department of Medicine, Miller School of Medicine, University of Miami, Miami, FL 33136, USA; 5Molecular Therapeutics Shared Resource, Sylvester Comprehensive Cancer Center, University of Miami, Miami, FL 33136, USA; 6Division of Hematology, Sylvester Comprehensive Cancer Center, University of Miami, Miami, FL 33136, USA

**Keywords:** AML, histone modifiers, HDAC inhibitors, BET inhibitors, drug screening, epigenetics

## Abstract

**Simple Summary:**

Acute myeloid leukemia is a cancer originating in the bone marrow and peripheral blood. Genetic mutations observed with the disease are difficult to target therapeutically and patients often have different treatment responses to the standard of care. By finding epigenetic compounds that work in a variety of patients, we could discover better therapies to treat the disease apart from the current standard of care. Here we present an unbiased drug screen of a variety of epigenetic compounds, with some showing effective responses in all or most patient samples.

**Abstract:**

The use of inhibitors of epigenetic modifiers in the treatment of acute myeloid leukemia (AML) has become increasingly appealing due to the highly epigenetic nature of the disease. We evaluated a library of 164 epigenetic compounds in a cohort of 9 heterogeneous AML patients using an ex vivo drug screen. AML blasts were isolated from bone marrow biopsies according to established protocols and treatment response to the epigenetic library was evaluated. We find that 11 histone deacetylase (HDAC) inhibitors, which act upon mechanisms of cell cycle arrest and apoptotic pathways through inhibition of zinc-dependent classes of HDACs, showed efficacy in all patient-derived samples. Other compounds, including bromodomain and extraterminal domain (BET) protein inhibitors, showed efficacy in most samples. Specifically, HDAC inhibitors are already clinically available and can be repurposed for use in AML. Results in this cohort of AML patient-derived samples reveal several epigenetic compounds with high anti-blast activity in all samples, despite the molecular diversity of the disease. These results further enforce the notion that AML is a predominantly epigenetic disease and that similar epigenetic mechanisms may underlie disease development and progression in all patients, despite differences in genetic mutations.

## 1. Introduction

Acute myeloid leukemia (AML) is a heterogeneous malignant disorder that arises from the clonal growth of myeloblasts in the bone marrow and peripheral blood. AML is characterized by a well-defined genetic landscape and a relatively low mutational burden [1]. The genomic alterations observed in AML patients, however, do not account for the heterogeneity observed clinically. Large-scale sequencing projects have revealed several potential AML driver genes, including *NPM1*, *CEBPA*, *DNMT3A*, *TET2*, *RUNX1*, *ASXL1*, *IDH2*, and *MLL*, and have identified critical mutations in *FLT3*, *IDH1*, *KIT*, and *RAS*. However, many patients have either driver mutations that cannot be directly clinically targeted or co-occurring mutations that drive resistance to available targeted therapies such as FLT3 and IDH inhibitors [2,3,4,5].

Recent studies have demonstrated the epigenetic nature of AML [6] and opened the door for therapies targeting epigenetic modifiers. Indeed, the majority of AML patients harbor somatic mutations in enzymes involved in either DNA methylation and histone modification and chromatin remodeling [7], and a hypermethylation pattern in the promoters of tumor suppressor genes is commonly observed [8]. While the role of aberrant methylation in leukemogenesis has been evaluated in depth [8,9], changes in histone acetylation and their impact on AML development are less clearly understood [10,11,12,13]. 

Histone acetylation is a dynamic process involved in the regulation of gene expression in response to intracellular and extracellular stimuli and is tightly controlled by the competing effects of histone lysine acetyltransferases (HATs) and histone deacetylases (HDACs). Histone acetylation generally leads to accessible chromatin structure facilitating gene transcription, while deacetylation results in gene silencing. Altered expression of both HATs and HDACs has been associated with abnormal DNA methylation and DNA hydroxymethylation and the silencing of tumor suppressor genes, a process facilitating malignant transformation [14]. DNA methyltransferase (DNMT) inhibitors such as 5-azacytidine (5-AZA) and 5-aza-2′deoxycytidine (decitabine) have been used for more than a decade in the treatment of AML patients with varying success rates and rapid development of treatment resistance [15]. HDAC inhibitors have been shown to upregulate the expression of regulatory genes involved in cell cycle arrest and apoptosis, such as p53, STAT, and MAPK/ERK, which are commonly silenced during cancer development and progression [11]. HDACs not only act upon histones, but they also have a variety of non-histone protein targets, including tumor suppressor genes and transcription factors [12]. Specific HDAC inhibitors have been shown to disrupt cancer cell proliferation through processes including interruption of DNA repair through non-homologous end-joining (NHEJ) [13], upregulation of pro-apoptotic Bcl2-member (BIM) [16], and degradation of FLT3 signaling in Akt, ERK, and STAT5 pathways. Although several HDAC inhibitors have been evaluated in clinical trials, none are clinically approved for myeloid neoplasms.

## 2. Material and Methods

### 2.1. Patient-Derived AML Samples

Patients with relapsed and refractory AML receiving standard-of-care bone marrow biopsies were recruited and written consent was obtained from all patients. All patients had exhausted standard-of-care treatment options at the time of enrollment. Bone marrow aspirates superfluous to clinical diagnosis were purified as described previously [16]. The study was approved by the University of Miami institutional review board (protocol numbers 20060858 and 20150989) and conducted in accordance with Good Clinical Practice guidelines. 

Bone marrow aspirates from 9 patients with relapsed/refractory AML were collected and the treatment response to a library of inhibitors of epigenetic modifiers was evaluated. Patients ranged in age from 26 to 83 years and were mixed with respect to ethnicity and gender, as prevalent in the south Florida catchment area. All patients had been treated and failed standard of care-intensive or non-intensive chemotherapy regimens. Apart from patient 7, all patients were analyzed by conventional G-band karyotyping, fluorescence in situ hybridization (FISH), and targeted sequencing for AML-relevant genes as part of the clinical routine. All patients previously failed or relapsed after receiving a standard-of-care chemotherapy regimen (3 + 7 chemotherapy consisting of the nucleoside analog cytarabine combined with a topoisomerase II inhibitor) and were exposed to multiple cycles of chemotherapy prior to recruitment.

### 2.2. Drug Screening

We tested a 164-compound epigenetic library that consists of inhibitors and activators of epigenetic modifying enzymes (writers, erasers, and readers) (Table 1). The epigenetics compound library contains compounds approved for the treatment of various cancers and diseases, tool compounds, and investigational compounds currently in preclinical or clinical development. Small molecule inhibitors targeting all major classes of histone modifiers are included (Histone deacetylase (HDAC) inhibitors (n = 51), Histone acetylase inhibitors (n = 6), Bromodomain and Extra-Terminal motif (BET) inhibitors (n = 11), Isocitrate dehydrogenase (IDH) inhibitors (n = 3), Methyltransferase inhibitors (n = 28), Demethylase inhibitors (n = 12), Sirtuin (SIRT) inhibitors (n = 11), poly ADP ribose polymerase (PARP) inhibitors (n = 3), p300/ CREB-binding protein (p300/CBP) inhibitors) (n = 3). Bone marrow aspirates were obtained, and mononuclear cells were isolated by Ficoll density gradient (Ficoll-Paque PREMIUM; Cytiva, Marlborough, MA, USA). Cells were washed, counted, and cultured in Mononuclear Cell Medium (MCM, PromoCell, Heidelberg, Germany) for 24 h. All stock compounds were dissolved in 100% DMSO and tested in duplicate at a nominal test concentration of 1 μM. Wells with assay buffer containing 0.1% DMSO served as negative controls. One thousand exponentially growing cells were seeded per well in 384-well micro-titer plates and incubated in the presence of compounds in a humidified environment at 37 °C and 5% CO_2_. After 72 h of treatment, cell viability was assessed by measuring ATP levels via bioluminescence (CellTiter-Glo, Promega, Madison, WI, USA). Positive hits were defined as any compound that showed cell killing higher than three standard deviations of the negative control. 

### 2.3. Drug Sensitivity Testing

Sixteen of the epigenetic compounds which showed activity in tested samples were chosen for drug sensitivity testing (DST) in two samples (Patient 8 and Patient 9), which was performed as described previously [16]. 

## 3. Results

### 3.1. Patients

Karyotyping and molecular analysis showed large genetic variation within the patient cohort that was representative of the general AML patient population. 

Patient 1 presented with AML characterized by t(8; 21) translocation (AML1-ETO), as well as *TET2, RAD21*, and *ETV6* mutations. Patient 2 presented with a history of prostate cancer treated with radiation therapy and AML with normal cytogenetics and *CEBPA* (monoallelic), *CSF3R*, and *TET2* mutations. Patient 3 displayed complex cytogenetics, including del(5q), trisomy 8, monosomy 7, deletion of chromosome 17, and a *TP53* missense mutation. Patient 4 had normal cytogenetics, four trisomies, including +8, and *NPM1*, *DNMT3A*, *KIT*, *SETBP1*, and *TET2* mutations. Patient 5 presented with complex cytogenetics, including del(5q), monosomy 3, and monosomy 12. Patient 6 presented with AML characterized by t(9;11) translocation [*KMT2A*]. Patient 8 had normal cytogenetics with mutations in *ASXL1*, *RUNX1, STAG2*, and *TET2*. Patient 9 presented with AML characterized by trisomy 8, mosaicism, 47, XY, +8 [14]/46, XY [6], and *BCOR*, *NRAS*, *SF3B1*, *TET2*, *DDX41*, *ETV6*, and *PDGFRA* mutations. Cytogenic data was not available for patient 7. All patients had relapsed/refractory disease after standard-of-care treatment, except patient 8, who did not receive treatment pre-biopsy.

### 3.2. Drug Sensitivity Profiles Differ between AML Samples

To identify epigenetic compounds that showed efficacy in this heterogeneous population of patient-derived AML samples, an ex vivo screen of 164 epigenetic compounds was performed. The epigenetics compound library contains a wide variety of FDA-approved and investigational compounds as well as tool compounds targeting epigenetic readers, writers, and erasers (Table 1).

The number of positive hits varied from 15 in sample 7 to 18 in sample 3, 19 in sample 5, 23 in sample 8, 26 in sample 6, 38 in sample 4, 40 in sample 1, and 58 positive hits in patient 9. Eleven compounds showed efficacy in all 9 patients and all of these were HDAC inhibitors. An additional four compounds showed efficacy in 8 of the 9 samples. These included 2 HDAC inhibitors and 2 BET inhibitors (Table 2, Figure 1).

While HDAC inhibitors represent the compound class with the most positive hits overall (n = 34), BET inhibitors are the second largest group of positive hits (n = 7). Of interest, although the use of methyltransferase inhibitors, such as Aza, is standard of care for AML [17], methyltransferase inhibitors did not have an effect in cell lines from 6 of the 9 patient-derived samples. Two methyltransferase inhibitors displayed activity in two samples, while five methyltransferase inhibitors displayed activity in only one sample (patient 9). This is likely as most patients received, and failed, standard-of-care regimens prior to the screen. Similarly, only a single demethylase inhibitor (JIB 04) displayed activity in more than one sample on the screen. Four additional demethylase inhibitors showed activity in a single sample (patient 9). One PARP inhibitor (OLAPAR 1B) displayed activity in 2 samples, while 3 SIRT inhibitors displayed activity in a single sample (patient 9). IDH inhibitors (n = 3), p300/CBP inhibitors (n = 3), and histone acetylase inhibitors (n = 6), displayed no activity in any of the evaluated samples.

Although this sample population displayed common responses to a subset of compounds, drug sensitivity profiles in response to the tested library differed between individual patients (Table 2).

Sample 9 had the highest number of positive hits candidates (n = 58), which included 26 HDAC inhibitors, 5 BET inhibitors, 7 methyltransferase inhibitors, 5 demethylase inhibitors, 3 SIRT inhibitors, 1 PARP inhibitor, and several other miscellaneous epigenetic inhibitors/activators. Twenty-one compounds showed efficacy in sample 9 only, including 5 methyltransferase inhibitors, 4 HDAC inhibitors, 4 demethylase inhibitors, 3 SIRT inhibitors, and 1 HAT activator. Sample 1 had the second highest number of positive hits candidates (n = 40), which included 27 HDAC inhibitors, 5 BET inhibitors, 1 methyltransferase inhibitor, and 1 demethylase inhibitor, amongst other miscellaneous epigenetic inhibitors. Four compounds showed efficacy in sample 1 alone, which were all HDAC inhibitors. Of the 28 compounds that worked in sample 2, 21 were HDAC inhibitors, and 4 were BET inhibitors. Sample 3 had 18 viable compounds, including 16 HDAC inhibitors, one histone demethylase inhibitor, and one proteasome inhibitor. Thirty-eight compounds showed efficacy in sample 4, including 25 HDAC inhibitors, 6 BET inhibitors, 1 methyltransferase inhibitor, 1 demethylase inhibitor, and several other miscellaneous inhibitors. One compound, a BET inhibitor (EP-336), showed efficacy in sample 4 alone. Sample 5 had 19 compound hits consisting of 13 HDAC inhibitors and 4 BET inhibitors. Sample 6 had 26 viable compound hits, including 17 HDAC inhibitors, 5 BET inhibitors, a demethylase inhibitor, and several miscellaneous inhibitors. Sample 7 had the least number of viable compound hits (15) and included 12 HDAC inhibitors, 2 BET inhibitors, and a dual PLK1 and BRD4 inhibitor. Sample 8 had 23 positive hits candidates, including 18 HDAC inhibitors, 3 BET inhibitors, 1 PARP inhibitor, and an adenosine kinase inhibitor. Two of the samples with the least number of hits, samples 3 and 5, present with complex karyotypes.

### 3.3. HDAC Inhibitors Display Activity in All AML Samples

Eleven of the tested compounds displayed activity in all seven samples. All 11 of these effective compounds were zinc-dependent HDAC inhibitors. Compounds that can be described as strong hits (i.e., ≥6 standard deviations) in six of the samples (Samples 1, 2, 4, 5, 6, and 9) include the HDAC inhibitors (S)-HDAC-42, Entinostat (MS-275), Quisinostat (JNJ-26481585), Dacinostat (LAQ824), NSC-3852, Belinostat (PXD101), and Trichostatin A. There were no active compounds (“strong hits”) when subject to a higher standard deviation in samples 2, 7, and 8.

A select number of HDAC (n = 11) and BET (n = 5) inhibitors were used to establish drug sensitivity profiles in two samples (Samples 8 and 9) to validate the single hit screen and evaluate sample-specific differences in drug efficacy. Results are represented as the Drug Sensitivity Score (DSS_mod_), which incorporates information on drug potency, efficacy, effect range, and therapeutic index. Compounds showed similar sensitivity scores and EC_50_ values between samples (Figure 2). Seven of the screened HDAC inhibitors displayed DSS_mod_ scores above 50 in both samples, including (S)-HDAC-42, Belinostat, Dacinostat, Fimepinostat, Quisinostat, Trapoxin A, and Trichostatin A. Quisinostat displayed the highest activity in both screened sample-derived cells with DSS_mod_ of 85.16 in sample 8 and 74.24 in sample 9. All screened BET inhibitors showed lower activities compared to the HDAC inhibitors displaying an EC_50_ between 0.21 nM (CPI203) and 2.48 nM (I-BET 762) (Figure 2).

Of the 11 HDAC inhibitors that showed efficacy in all 9 samples, 3 compounds have completed or are currently undergoing Phase I and II clinical trials in AML patients (Table 3), while two of the tested compounds that showed efficacy in some of the patients (SAHA and SB 939) have reached Phase III trials for AML.

HDAC inhibitors are a relatively novel class of compounds. Of the 11 HDAC inhibitors that showed efficacy in all 9 samples, the specific HDAC targets and potency, i.e., IC_50_ values, are only known for some. There is a lack of consensus on the selectivity and IC_50_ of each of these compounds towards HDAC enzymes, making it difficult to determine the commonality between the 11 HDAC inhibitors. Eight of the compounds have reported enzymatic targets, and of those 8 compounds, all are reported to inhibit HDAC1 (Table 4). However, many of the other HDAC inhibitors that did not show efficacy in all patients also inhibit HDAC1. HDAC2 and HDAC6 are also common targets of many of the compounds.

Trichostatin A, Belinostat (PXD101), and Quisinostat (JNJ-26481585) are all structurally related to the commonly known pan-HDAC inhibitor SAHA, with slightly differing isoform selectivity. Trichostatin A and Belinostat (PXD101) are all reported to have the highest affinities for HDACs 1, 2, 3, and 6, with different selectivity for other class I and II HDACs [33]. Quisinostat (JNJ-2648158) is reported to be most selective for HDACs 1, 2, 4, 10, and 11 but also inhibits other zinc-dependent HDACs [33]. Entinostat (MS-275) is a member of the *ortho*-aminoanilide family of HDAC inhibitors and acts selectively on Class I HDACs 1, 2, and 3 [33]. Dacinostat (LAQ824) is known to inhibit HDAC1 but is also reported as a potent HDAC inhibitor and was shown to be effective against myeloid leukemia cells in vitro and in vivo [34]. Trapoxin A is reported to be an irreversible inhibitor of class I HDACs but may act on other classes of HDACs as well [35]. LMK-235 most selectively inhibits HDAC4 and HDAC5 but also acts on other zinc-dependent HDACs [36]. Fimepinostat (CUDC-907) potently inhibits class I and II HDACs and also inhibits class I phosphoinositide 3-kinases (PI3Ks), which are involved in cancer cell proliferation and survival [37]. M-344 targets HDACs 1, 2, 3, 6, 8, and 10 [38]. The specific HDAC targets of Oxamflatin [39], (S)-HDAC-42 (AR-42) [40], and NSC-3852 [41] are not currently known.

Out of the select compounds that showed efficacy in our screen, only a handful are currently in clinical trials (Table 3), potentially supporting the future development of these compounds clinically.

## 4. Discussion

The standard-of-care backbone for younger patients with AML using the combination of the nucleoside analog cytarabine and an anthracycline has changed little over the last decade, with the exception of adding midostaurin in *FLT3*-mutated patients, and is only moderately effective in the majority of patients (~40–50% 5-year OS) [42]. In older or unfit patients, HMA-based therapy, typically in combination with venetoclax, has emerged as the new standard of care over the last five years, with variable success (median OS ~15 months) [43,44]. The limited success of these treatment regimens has been attributed to high levels of heterogeneity found in these patients with respect to disease progression and treatment response, a phenotype that is further attenuated in patients with relapsed/refractory disease. Indeed, we have recently shown that patients with relapsed/refractory AML display vastly different drug sensitivity profiles in response to treatment with a library of 215 FDA-approved compounds [16], with no single compound exhibiting activity above the threshold in all patients. In that study, clinically relevant mutations failed to predict treatment responses, similar to what has been observed clinically.

An increased understanding of the impact of epigenetic dysregulation on AML development and progression has resulted in the development of novel therapeutics targeting epigenetic modifiers.

Aberrant DNA methylation has been described as an essential step in AML development and progression, in part due to a high rate of mutations in *DNMT3A* and *TET2* [2,45]. Single-agent DNMT inhibitors, such as 5-azacytidine (5-AZA) and 5-aza-2′deoxycytidine (decitabine, DAC), have been used to treat AML patients. While these compounds displayed promising results in clinical trials, treatment failure is observed in most patients [46,47].

Despite the promising pre-clinical activity, single-agent HDAC1/2 inhibitors have had limited clinical success to date in myeloid neoplasms. Although no mutations in HDAC genes have been described, HDAC proteins have been shown to be aberrantly recruited to specific gene promoters [48]. The chimeric fusion protein AML1-ETO, for example, recruits HDAC1, HDAC2, and HDAC3, thereby silencing AML1 target genes, which results in differentiation arrest and leukemic transformation [48]. Although a number of HDAC inhibitors are currently being evaluated preclinically and in clinical trials, real-world evidence of clinical success is lacking [49]. This may be because 5-AZA is not the best combination partner for use with an HDAC inhibitor in myelodysplastic syndrome (MDS) or AML. 5-AZA’s anti-tumor activity is dependent on cell cycling, and HDAC inhibitors down-regulate or arrest the cell cycle.

We have evaluated a large and diverse library of 164 inhibitors of epigenetic modifiers in a small cohort of samples derived from patients with relapsed/refractory AML. In contrast to our previous observation in a cohort of relapsed/refractory AML patients where drug sensitivity profiles differed widely between patients in response to treatment with FDA-approved agents, our results showed a different pattern in response to treatment with the epigenetic library, where a few HDAC inhibitors displayed activity in all of the samples, irrespective of cytogenetic or molecular drivers. HDAC inhibitors were the only class of epigenetic compounds that displayed activity in all samples. Although the specificity towards HDAC family members has not been described for all the compounds, most of them display a common affinity towards HDAC1, suggesting an underlying dependency on histone or protein acetylation that needs to be investigated.

Additionally, 11 of the HDAC inhibiting compounds we selected to perform a DST screen validated the efficacy of these compounds against our AML blasts. Both of the tested samples responded well to these HDAC inhibitors, although at varying multitudes, which is expected given the heterogeneity of the disease. Our data add to the large body of preclinical evidence supporting the further development of HDAC inhibitors in AML. While many of the compounds represented in the screening library are indeed tool compounds, the activity profiles can be used as a foundation for future drug development efforts and suggest efficacious combination therapies. As new compounds are being continuously developed and evaluated first as monotherapy and then in combination with established treatment regimens, novel chemotherapy or venetoclax-based regimens that include an HDAC inhibitor may enhance efficacy, particularly in the relapsed/refractory setting where resistance to standard agents is high.

BET inhibitors have shown promise in preclinical and clinical studies in a number of malignancies and are currently being investigated for use in AML in early-phase clinical trials [50]. Patients with myeloid malignancies that harbor ASXL1 mutations may be selectively sensitive to BET inhibitors [51]. This compound class represented the second largest number of effective compounds in this sample cohort, validating our screening results. Although most of these compounds are either tool compounds or compounds in the early stages of pre-clinical and clinical development, our results further support the clinical exploration of this compound class.

Notably, our screen identified only one compound that inhibits DNMT from being effective in a single sample, suggesting that these aberrations may be patient-specific. It has been demonstrated, however, that HDAC inhibition can alter DNMT levels and, subsequently, DNA methylation [14].

Despite the genetic diversity observed in the sample cohort, our results suggest a potential role for epigenetic aberrations, specifically in histone acetylation, in AML disease progression. Novel, more selective HDAC inhibitors and novel HDAC combination therapies should be further explored in myeloid malignancies.

## 5. Conclusions

In summary, HDAC inhibiting compounds showed the most widespread efficacy in our AML sample population despite the heterogeneity of their cytogenic and molecular signatures. Our data support the rationale for the further assessment of epigenetic compounds, especially HDAC inhibitors, for AML therapeutics.

## Figures and Tables

**Figure 1 cancers-14-04094-f001:**
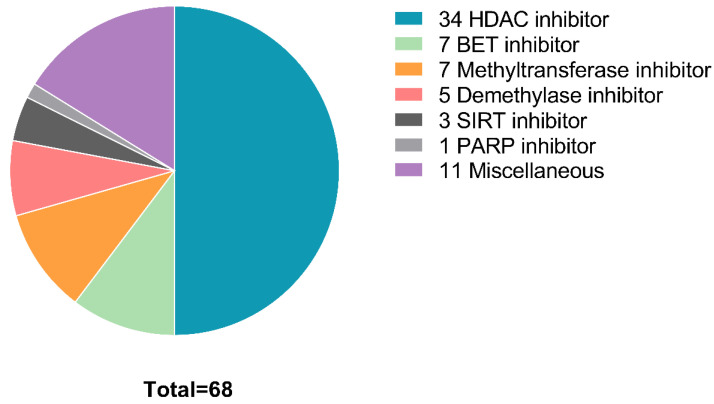
Distribution of compound types that showed efficacy in at least one sample. BET, bromodomain and extra-terminal motif; HDAC, histone deacetylase; SIRT, surtuin; PARP, poly (ADP-ribose) polymerase.

**Figure 2 cancers-14-04094-f002:**
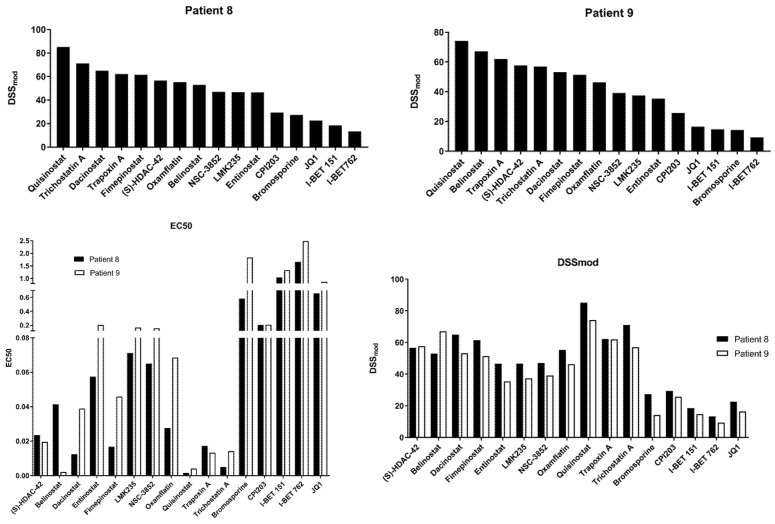
EC_50_ and drug sensitivity scoring for 16 select compounds in two patient-derived cell lines.

**Table 1 cancers-14-04094-t001:** Compound list of the Epigenetics modifier library.

Class	Compounds
**HDAC inhibitors (n = 51)**	(S)-HDAC-42, ACY-1215 (Rocilinostat), Apicidin, BATCP, BML-210, BML-281, CAY10398, CAY10603, CBHA, Chidamide, CI-994, CUDC-101, CUDC-907, Droxinostat, Entinostat (MS-275), Fluoro-SAHA, Givinostat (ITF2357), ITSA-1, JNJ-26481585, KD 5170, LAQ824, LMK 235, M-344, MC 1568, MC-1293, NCH-51, Nexturastat A, NSC-3852, Nullscript, Oxamflatin, PCI 34051, PCI-24781 (Abexinostat), Phenylbutyrate·Na, PXD101, Pyroxamide, RG2833 (RGFP109), RGFP966, SAHA, SB 939, SBHA, Scriptaid, Sodium 4-Phenylbutyrate, Suberoyl bis-hydroxamic acid, TC-H 106, TCS HDAC6 20b, TMP269, Trapoxin A, Trichostatin A, Tubastatin A, Valproic acid, Valproic acid hydroxamate
**Histone acetylase inhibitors (n = 6)**	Butyrolactone 3, CPTH2, Delphinidin chloride, Garcinol, MB-3, NU 9056
**BET inhibitors (n = 11)**	(+)-JQ1, Bromosporine, CPI203, EP-313, EP-336, GSK2801, I-BET 151, I-BET762 (GSK525762), PFI 1, PFI-3, RVX-208
**IDH inhibitor (n = 3)**	(R)-2-HG, AGI-5198 (IDH-C35), AGI-6780,
**Methyltransferase inhibitors (n = 28)**	(R)PFI-2, 2′-Deoxy-5-fluorocytidine, 5-Aza-2′-deoxycytidine, A-366, BIX-01294·3HCl, DZNep, Entacapone, EPZ005687, EPZ-5676, EPZ-6438, GSK126, GSK343, LLY-507, Lomeguatrib, MM-102, RG 108, SGC0946, SGI-1027, Sinefungin, UNC 0224, UNC 0638, UNC 0646, UNC 926, UNC0321, UNC0642, UNC1215, UNC1999, UNC669, Zebularine
**Demethylase inhibitors (n = 12)**	2,4-Pyridinedicarboxylic Acid, GSK-J1, GSK-J2, GSK-J4, GSK-J5, IOX 1, IOX 2, JIB 04, OG-L002, PBIT, RN-1, Tranylcypromine hemisulfate
**Sirt inhibitor (n = 11)**	AGK2, AK-7, BML-266, CAY10591, EX-527, Nicotinamide, Salermide, Sirtinol, Splitomicin, SRT1720, Tenovin-1
**PARP inhibitors (n = 3)**	BYK 204165, OLAPAR 1B, PJ 34
**P300/CBP inhibitors (n = 3)**	C 646, I-CBP 112, SGC-CBP30
**Miscellaneous activators and inhibitors (n = 35)**	5-Iodotubercidin, 6-Thioguanine, Aminoresveratrol sulfate, Anacardic acid, APHA, B2, BI-2536, BML-278, Cl-Amidine, CTPB, Curcumin, Daminozide, Disulfiram, Ebselen, Ellagic Acid, Hydralazine, Isonicotinamide, LSD1-C12, LSD1-C76, LY294002, MI-2, P22077, Piceatannol, Plumbagin, PTC-209, Resveratrol, SBI-7406, SBI-7673, SBI-8162, Suramin·6Na, TG101348 (SAR302503), Triacetylresveratrol, UPF 1069, WDR5-C47, β-Lapachone

Compounds are ordered based on the mechanism of action.

**Table 2 cancers-14-04094-t002:** Treatment responses in individual patients.

Samples	Hit Compounds
**Sample 1**	(+)-JQ1, (S)-HDAC-42 (AR-42), 2′-Deoxy-5-fluorocytidine, 5-Iodotubercidin, ACY-1215 (Rocilinostat), Apicidin, BI-2536, BML-281, Bromosporine, CAY10603, CBHA, Chidamide, CPI203, CUDC-101, CUDC-907 (Fimepinostat), Disulfiram, Entinostat (MS-275), Fluoro-SAHA, Givinostat (ITF2357), I-BET 151, I-BET62 (GSK525762), JIB 04, JNJ-26481585, KD 5170, LAQ824 (Dacinostat), LMK 235, LSD1-C12, M-344, NSC-3852, Oxamflatin, PXD101 (Belinostat), Pyroxamide, SAHA, SB 939 (Pracinostat), Scriptaid, Suberoyl bis-hydroxamic acid, TG101348 (SAR302503), Trapoxin A, Trichostatin A, β-Lapachone
**Sample 2**	(+)-JQ1, (S)-HDAC-42 (AR-42), 5-Iodotubercidin, Apicidin, BI-2536, CBHA, CPI203, CUDC-907 (Fimepinostat), Entinostat (MS-275), Fluoro-SAHA, Givinostat (ITF2357), I-BET 151, I-BET62 (GSK525762), JNJ-26481585, LAQ824 (Dacinostat), LMK 235, M-344, Nexturastat A, NSC-3852, Oxamflatin, PCI-24781 (Abexinostat), PXD101 (Belinostat), SAHA, SB 939 (Pracinostat), Scriptaid, Trapoxin A, Trichostatin A, β-Lapachone
**Sample 3**	(S)-HDAC-42 (AR-42), CUDC-907 (Fimepinostat), Disulfiram, Entinostat (MS-275), Givinostat (ITF2357), JIB 04, JNJ-26481585, LAQ824 (Dacinostat), LMK 235, M-344, NSC-3852, Oxamflatin, PCI-24781 (Abexinostat), PXD101 (Belinostat), SAHA, SB 939 (Pracinostat), Trapoxin A, Trichostatin A
**Sample 4**	(+)-JQ1, (S)-HDAC-42 (AR-42), 5-Iodotubercidin, ACY-1215 (Rocilinostat), Apicidin, BI-2536, Bromosporine, CBHA, Chidamide, CI-994, CPI203, CUDC-101, CUDC-907 (Fimepinostat), Disulfiram, DZNep, Entinostat (MS-275), EP-336, Fluoro-SAHA, Givinostat (ITF2357), I-BET 151, I-BET62 (GSK525762), JIB 04, JNJ-26481585, LAQ824 (Dacinostat), LMK 235, M-344, Nexturastat A, NSC-3852, Oxamflatin, PCI-24781 (Abexinostat), PXD101 (Belinostat), SAHA, SB 939 (Pracinostat), Scriptaid, TG101348 (SAR302503), Trapoxin A, Trichostatin A, β-Lapachone
**Sample 5**	(+)-JQ1, (S)-HDAC-42 (AR-42), BI-2536, CPI203, CUDC-907 (Fimepinostat), Disulfiram, Entinostat (MS-275), Givinostat (ITF2357), I-BET62 (GSK525762), JNJ-26481585, LAQ824 (Dacinostat), LMK 235, NSC-3852, Oxamflatin, PXD101 (Belinostat), SB 939 (Pracinostat), Trapoxin A, Trichostatin A
**Sample 6**	(+)-JQ1, (S)-HDAC-42 (AR-42), 5-Iodotubercidin, Apicidin, BI-2536, CPI203, CUDC-907, Disulfiram, Entinostat (MS-275), EP670, Fluoro-SAHA, Givinostat (ITF2357), -BET 151, I-BET62 (GSK525762), JIB 04, JNJ-26481585, LAQ824 (Dacinostat), LMK 235, M-344, NSC-3852, Oxamflatin, PXD101 (Belinostat), SB 939 (Pracinostat), Scriptaid, Trapoxin A, Trichostatin A
**Sample 7**	(+)-JQ1, (S)-HDAC-42 (AR-42), Apicidin, BI-2536, CPI203, CUDC-907, Entinostat (MS-275), JNJ-26481585, LAQ824 (Dacinostat), LMK 235, M-344, NSC-3852, Oxamflatin, PXD101 (Belinostat), Trapoxin A, Trichostatin A
**Sample 8**	(+)-JQ1, (S)-HDAC-42 (AR-42), 5-Iodotubercidin, Apicidin, CBHA, CPI203, CUDC-907, Entinostat (MS-275), EP670, Givinostat (ITF2357), JNJ-26481585, LAQ824, LMK235, M-344, Nexturastat A, NSC-3852, OLAPAR 1B, Oxamflatin, PXD101 (Belinostat), SB 939 (Pracinostat), Scriptaid, Trapoxin A, Trichostatin A
**Sample 9**	(+)-JQ1, (S)-HDAC-42 (AR-42), 2′-Deoxy-5-fluorocytidine, 5-Iodotubercidin, AK-7, Apicidin, BI-2536, BML-266, CBHA, CI-994, CPI203, CTPB, CUDC-907, Disulfiram, DZNep, Entinostat (MS-275), EP670, EPZ-5676, Fluoro-SAHA, Givinostat (ITF2357), GSK126, GSK-J1, I-BET 151, I-BET762 (GSK525762), JIB 04, JNJ-26481585, LAQ824, LMK 235, LSD1-C12, LY294002, M-344, MM-102, Nexturastat A, Nicotinamide, NSC-3852, OG-L002, OLAPAR 1B, Oxamflatin, P22077, PBIT, PTC-209, Piceatannol, PXD101 (Belinostat), Pyroxamide, RG 108, RGFP966, RN-1, SAHA, SB 939 (Pracinostat), SBHA, SBI-7673, Scriptaid, SGI-1027, TC-H 106, TCS HDAC6 20b, TG101348 (SAR302503), Trapoxin A, Trichostatin A

**Table 3 cancers-14-04094-t003:** Efficacious epigenetic compounds in clinical evaluation for AML.

Compound	*N*	Mode of Action	Clinical Trial for AML
**2′-Deoxy-5-fluorocytidine**	2	DNMT inhibitor	Phase I [18]
**Abexinostat (PCI-24781)**	3	HDAC inhibitor	Phase I [19]
**Belinostat (PXD101)**	9	HDAC inhibitor	Phase I, Phase II [20]
**BI-2536**	7	PLK and BRD4 inhibitor	Phase I/II [21]
**Chidamide**	2	HDAC inhibitor	Phase I and II
**Entinostat (MS-275)**	9	HDAC inhibitor	Phase I [22], Phase II [23]
**Fedratinib (TG101348/SAR302503)**	3	JAK2-selective inhibitor	Phase I [24], Phase II
**Pracinostat (SB 939)**	8	HDAC inhibitor	Phase I [25], Phase II [26], Phase III [27]
**Pyroxamide**	2	HDAC inhibitor	Phase I
**SAHA (Vorinostat)**	5	HDAC inhibitor	Phase I [28,29]; Phase II [30,31,32]; Phase III
**Trichostatin A**	9	HDAC inhibitor	Phase I

**Table 4 cancers-14-04094-t004:** Known HDAC targets for HDAC inhibitors that showed efficacy in all patients.

Compounds	Class I				Class IIa				Class IIb		Class IV
	HDAC1	HDAC2	HDAC3	HDAC8	HDAC4	HDAC5	HDAC7	HDAC9	HDAC6	HDAC10	HDAC11
(S)-HDAC-42	UNKNOWN										
CUDC-907 (Fimepinostat)	**X**	**X**	**X**	**X**	**X**	**X**	**X**	**X**	**X**	**X**	**X**
Entinostat (MS-275)	**X**	**X**	**X**								
JNJ-26481585 (Quisinostat)	**X**	**X**	**X**	**X**	**X**	**X**	**X**	**X**	**X**	**X**	
LAQ824 (Dacinostat)	**X**										
LMK 235	**X**	**X**		**X**	**X**	**X**			**X**		**X**
NSC-3852	UNKNOWN										
OXamflatin	UNKNOWN										
PXD101 (Belinostat)	**X**	**X**	**X**						**X**		
TrapoXin A	**X**	**X**	**X**	**X**							
Trichostatin A	**X**	**X**	**X**						**X**		

## Data Availability

Data is contained within the article.

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
