# Peer review of "Functional Drug Screening of Small Molecule Inhibitors of Epigenetic Modifiers in Refractory AML Patients"

_cancers, 2022, doi:10.3390/cancers14174094_

Round 1

Reviewer 1 Report

The manuscript presents a small-scale screen of epigenetic modifiers in leukemic blasts of Acute Myeloid Leukemia patients. Overall, the manuscript is well organized, structured and the message flows very well throughout the text. However, the data presented is quite simplistic (despite the amount of research required for it), a pilot screen showing that several epigenetic modifiers impact on the viability of myeloid leukemia blasts, no glimpse on the possible molecular mechanisms that might explain the functional effects of the pharmacological agents. Importantly, the title of the manuscript is also misleading, Histone Deacetylase inhibitors are enriched in the initial library (51/164), therefore it is no surprise that are also more represented in the screen outcome.

Some minor aspects that were also noted:

- Table 4 is out of format

-Line 250 – it should be Table 3 and not Table 4.

Author Response

Reviewer 1:

The manuscript presents a small-scale screen of epigenetic modifiers in leukemic blasts of Acute Myeloid Leukemia patients. Overall, the manuscript is well organized, structured and the message flows very well throughout the text. However, the data presented is quite simplistic (despite the amount of research required for it), a pilot screen showing that several epigenetic modifiers impact on the viability of myeloid leukemia blasts, no glimpse on the possible molecular mechanisms that might explain the functional effects of the pharmacological agents. Importantly, the title of the manuscript is also misleading, Histone Deacetylase inhibitors are enriched in the initial library (51/164), therefore it is no surprise that are also more represented in the screen outcome.

We have changed the title to make this clear.

Some minor aspects that were also noted:     

- Table 4 is out of format

This was corrected

-Line 250 – it should be Table 3 and not Table 4.

This was corrected.

Reviewer 2 Report

The manuscript “Functional Drug Screening of Small Molecule Inhibitors of Epigenetic Modifiers in Refractory AML Patients Shows Enrichment for Histone Deacetylase Inhibitors” explored a library of 164 epigenetic compounds in a cohort of 9 heterogeneous AML patients’ samples. AML blasts were isolated from patients. The authors found out that 11 histone deacetylase (HDAC) inhibitors took effect in all patient samples, bromodomain and extraterminal domain (BET) protein inhibitors were affected in most patients. Overall, this manuscript is quite interesting and of great clinical significance. Besides, all the methods are performed reasonably.  I do recommend acceptance of this manuscript for publication in the journal.

Author Response

Reviewer 2:

The manuscript “Functional Drug Screening of Small Molecule Inhibitors of Epigenetic Modifiers in Refractory AML Patients Shows Enrichment for Histone Deacetylase Inhibitors” explored a library of 164 epigenetic compounds in a cohort of 9 heterogeneous AML patients’ samples. AML blasts were isolated from patients. The authors found out that 11 histone deacetylase (HDAC) inhibitors took effect in all patient samples, bromodomain and extraterminal domain (BET) protein inhibitors were affected in most patients. Overall, this manuscript is quite interesting and of great clinical significance. Besides, all the methods are performed reasonably.  I do recommend acceptance of this manuscript for publication in the journal.

Thank you for your evaluation.

Reviewer 3 Report

In their article, Dennison et al addressed the issue of ex vivo sensitivity to drugs characterized by an effect on epigenetics in relapsed/refractory AML. In spite of some interesting points that are raised by the authors, the article has several major issues to be addressed.

Major issues:

In the introduction, the authors state “however, the majority of mutations cannot be clinically targeted” (line 44). This statement sounds incorrect as the modern therapy of AML is progressively incorporating (even if with variable outcome) targeted inhibitors for a conspicuous number of the quoted genes (FLT3, IDH, MLL, KIT). As such, this rationale for investigating epigenetic modifiers turns out to be backward.

In the introduction section, the authors state “Although several HDAC inhibitors have been evaluated in clinical trials, none are clinically approved for myeloid neoplasms” (lines 72-73). That poses a major issue for the rationale of the present study: the sentence betrays a scarce efficacy by such therapeutic approach and the authors should discuss here the results of the main clinical trials and why in spite of disappointing clinical results, they decided to investigate this pharmacologic category in refractory AML.

Patients: the study recruited 9 patients with relapsed/refractory disease. Although methodologically interesting, that is definitely a very limited study cohort, even more if we consider the wide heterogeneity that characterizes R/R AML in general and in the study cohort specifically, as clearly emerges from the genetic features of patients at baseline (section 3.1). As an example above the others, the age ranges from 26 to 83; it is well-established the prognostic role of age and the difference in therapeutic approach in age strata, that clearly limits the potential applicability of the proposed approach.

Patients: the authors state “all the patients received standard-of care chemotherapy […] and were exposed to multiple rounds of chemotherapy”. I would ask if the 86-y-old patient received standard chemotherapy also because usually at such an age, chemotherapy is not considered a standard as a patient would be considered intrinsically (by age) unfit. Also, “rounds of chemotherapy” is not a proper nomenclature, it would be better to rephrase to “cycles”.

Results: the number of compounds are different in the different types, in particular HDAC inhibitors are significantly more than the other categories. As such, the distribution of the effective compounds is not balanced and the enrichment in HDACi could be the results of the higher opportunity to find an effect among a larger number of agents. The authors should normalize this analysis in order to clean from confounding effects.

Discussion: the authors state “The standard-of-care backbone for patients with AML using the combination of the nucleoside analogue cytarabine and an anthracycline has changed little over the last decade and is only moderately effective in the majority of patients”. That is definitely incorrect. This statement could be related to the fact that the authors quote a review published in 2016 (De Kouchkovsky I, Abdul-Hay M. ‘Acute myeloid leukemia: a comprehensive review and 2016 update.’ Blood Cancer J. 409 2016) whereas in the last five years the clinical scenario of AML has significantly changed. I strongly encourage the authors to evaluate and quote more updated references in their discussion.

Discussion: the authors state “Histone acetylation has only recently been evaluated in AML”. That is incorrect. Several clinical trials have been conducted in myeloid neoplasms and also AML demonstrating scarce clinical effectiveness in monotherapy (as well-discussed in a recent review in Cancers by Edurne San José-Enériz, 2019, that the authors have quoted), although having shown promising results in preclinical AML models.

Conclusions: for all the above points, the conclusions (“our results suggest a major role for epigenetic aberrations, specifically in histone acetylation, in AML disease progression”) are definitely an overstatement.

Minor issues:

The names of genes should be reported in italics.

Author Response

Reviewer 3:

In their article, Dennison et al addressed the issue of ex vivo sensitivity to drugs characterized by an effect on epigenetics in relapsed/refractory AML. In spite of some interesting points that are raised by the authors, the article has several major issues to be addressed.

Major issues:

In the introduction, the authors state “however, the majority of mutations cannot be clinically targeted” (line 44). This statement sounds incorrect as the modern therapy of AML is progressively incorporating (even if with variable outcome) targeted inhibitors for a conspicuous number of the quoted genes (FLT3, IDH, MLL, KIT). As such, this rationale for investigating epigenetic modifiers turns out to be backward.

Thank you for this comment and you are correct.  We have modified that sentence to state:

“however, many patients have either driver mutations that cannot be directly clinically targeted, or co-occurring mutations that drive resistance to available targeted therapies such as FLT3 and IDH inhibitors”.

We also note that only FLT3 and IDH inhibitors currently have regulatory approval for clinical use with, as you suggest, variable outcomes in patients.  We believe epigenetic therapies still have an important role to play in AML, potentially in combination with mutation-targeted therapies.

In the introduction section, the authors state “Although several HDAC inhibitors have been evaluated in clinical trials, none are clinically approved for myeloid neoplasms” (lines 72-73). That poses a major issue for the rationale of the present study: the sentence betrays a scarce efficacy by such therapeutic approach and the authors should discuss here the results of the main clinical trials and why in spite of disappointing clinical results, they decided to investigate this pharmacologic category in refractory AML.

Thank you for this important comment.  We note in the Discussion that one potential limitation of prior HDAC trials in MDS/AML was the choice of combination partner – in many cases azacitidine – which requires active cell cycling for clinical effect, which HDAC inhibitors abrogate.  In addition, pan-HDAC inhibitors in many cases prove too toxic in MDS and AML, which may have prevented or limited the use of clinically efficacious doses.  More targeted, selective HDAC inhibitors are under development and our data suggest that these may still have a viable role in myeloid neoplasms. 

Patients: the study recruited 9 patients with relapsed/refractory disease. Although methodologically interesting, that is definitely a very limited study cohort, even more if we consider the wide heterogeneity that characterizes R/R AML in general and in the study cohort specifically, as clearly emerges from the genetic features of patients at baseline (section 3.1). As an example above the others, the age ranges from 26 to 83; it is well-established the prognostic role of age and the difference in therapeutic approach in age strata, that clearly limits the potential applicability of the proposed approach.

We agree that this is a limitation of studying R/R AML, although this remains an area of extremely high unmet need.  Our study was an exploratory, proof-of-concept study of the platform and technology, focused on epigenetic targeted therapies.

Patients: the authors state “all the patients received standard-of care chemotherapy […] and were exposed to multiple rounds of chemotherapy”. I would ask if the 86-y-old patient received standard chemotherapy also because usually at such an age, chemotherapy is not considered a standard as a patient would be considered intrinsically (by age) unfit. Also, “rounds of chemotherapy” is not a proper nomenclature, it would be better to rephrase to “cycles”.

Thank you, we have modified this to say, “standard of care intensive or non-intensive chemotherapy” and changed “rounds” to “cycles”.

Results: the number of compounds are different in the different types, in particular HDAC inhibitors are significantly more than the other categories. As such, the distribution of the effective compounds is not balanced and the enrichment in HDACi could be the results of the higher opportunity to find an effect among a larger number of agents. The authors should normalize this analysis in order to clean from confounding effects.

The data shown in this manuscript represents unbiased screens. Meaning that cancer cells either die in response to drug treatment or they don’t. There is no confounding bias/effect that can account for the cells dying more efficiently in response to treatment with a specific HDAC inhibitor just because more of these compounds are present on the library. We can therefore no normalize the data.

The library is enriched for HDAC inhibitors simply because more of these compounds are commercially available than for example BET inhibitors. The library contains epigenetic modifiers of different classes that are available for us.

Discussion: the authors state “The standard-of-care backbone for patients with AML using the combination of the nucleoside analogue cytarabine and an anthracycline has changed little over the last decade and is only moderately effective in the majority of patients”. That is definitely incorrect. This statement could be related to the fact that the authors quote a review published in 2016 (De Kouchkovsky I, Abdul-Hay M. ‘Acute myeloid leukemia: a comprehensive review and 2016 update.’ Blood Cancer J. 409 2016) whereas in the last five years the clinical scenario of AML has significantly changed. I strongly encourage the authors to evaluate and quote more updated references in their discussion.

Thank you for this important point.  We have changed this to, “The standard-of-care backbone for younger patients with AML using the combination of the nucleoside analogue cytarabine and an anthracycline has changed little over the last decade, with the exception of adding midostaurin in FLT3-mutated patients, and is only moderately effective in the majority of patients (~40-50% 5-year OS)” [42, 43].  “In older or unfit patients, HMA-based therapy, typically in combination with venetoclax, has emerged as the new standard of care over the last 5 years, with variable success (median OS ~15 months) [44].”

We have added another reference: Stone et al, NEJM 2017.

Discussion: the authors state “Histone acetylation has only recently been evaluated in AML”. That is incorrect. Several clinical trials have been conducted in myeloid neoplasms and also AML demonstrating scarce clinical effectiveness in monotherapy (as well-discussed in a recent review in Cancers by Edurne San José-Enériz, 2019, that the authors have quoted), although having shown promising results in preclinical AML models.

Thank you.  We have changed this sentence to “Despite promising pre-clinical activity, single agent HDAC1/2 inhibitors have had limited clinical success to date in myeloid neoplasms…”.

Conclusions: for all the above points, the conclusions (“our results suggest a major role for epigenetic aberrations, specifically in histone acetylation, in AML disease progression”) are definitely an overstatement.

Thank you.  We have changed “major” to “potential”.  Also, we added “…in AML disease progression.  Novel, more selective HDAC inhibitors and/or novel HDAC combination therapies should be further explored in myeloid malignancies.”

Minor issues:

The names of genes should be reported in italics.

This was corrected.

Reviewer 4 Report

Dennison et al. examined the treatment response to a library of inhibitors of epigenetic modifiers using bone marrow aspirates from nine patients with relapsed/refractory AML. The authors evaluated a 164-compound epigenetic library containing inhibitors and activators of enzymes that modify epigenetic information. Sixteen of the epigenetic compounds that demonstrated activity in test samples were selected for testing the drug sensitivity of patient samples. Sixty-eight types of compounds demonstrated efficacy in at least one patient, with HDAC inhibitors representing the class of compounds with the highest number of positive hits. Eleven of the tested substances were active in all seven patients. All eleven of these effective substances were zinc-dependent HDAC inhibitors. In the majority of tested samples, the authors identified a common affinity for HDAC1, indicating the need to investigate an underlying dependence on histone or protein acetylation. These data are, in my opinion, preclinical evidence supporting the continued development of HDAC inhibitors for AML. My additional comments are listed below.

Paper and abstract refer patient samples as “patient” or “response in patients”, to me this is misleading because observed responses are seen in the samples, not in patients

line 299 – “Both of the tested patients responded well to these HDAC inhibitors, although at varying multitudes, which is expected given the heterogeneity of the disease”

Abstract  - “Other compounds including bromodomain and extraterminal domain (BET) protein 27 inhibitors showed efficacy in most patients

Abstract:  “ant-blast activity” typo? 

Author Response

Reviewer 4:

Dennison et al. examined the treatment response to a library of inhibitors of epigenetic modifiers using bone marrow aspirates from nine patients with relapsed/refractory AML. The authors evaluated a 164-compound epigenetic library containing inhibitors and activators of enzymes that modify epigenetic information. Sixteen of the epigenetic compounds that demonstrated activity in test samples were selected for testing the drug sensitivity of patient samples. Sixty-eight types of compounds demonstrated efficacy in at least one patient, with HDAC inhibitors representing the class of compounds with the highest number of positive hits. Eleven of the tested substances were active in all seven patients. All eleven of these effective substances were zinc-dependent HDAC inhibitors. In the majority of tested samples, the authors identified a common affinity for HDAC1, indicating the need to investigate an underlying dependence on histone or protein acetylation. These data are, in my opinion, preclinical evidence supporting the continued development of HDAC inhibitors for AML. My additional comments are listed below.

Paper and abstract refer patient samples as “patient” or “response in patients”, to me this is misleading because observed responses are seen in the samples, not in patients

This was corrected in the manuscript.

line 299 – “Both of the tested patients responded well to these HDAC inhibitors, although at varying multitudes, which is expected given the heterogeneity of the disease”

This was corrected in the manuscript.

Abstract  - “Other compounds including bromodomain and extraterminal domain (BET) protein 27 inhibitors showed efficacy in most patients

This was corrected in the manuscript.

Abstract:  “ant-blast activity” typo? 

This was corrected in the manuscript.

Round 2

Reviewer 1 Report

The authors addressed most of the concerns raised in the previous revision.

Reviewer 3 Report

The authors addressed several of the raised points.

In my opinion the work remains limited by some major issues, particularly the sample size (in spite of acknowledging that they have not increased their study population) and heterogeneity of the studied patients.